# Chromoblastomycosis in an Endemic Area of Brazil: A Clinical-Epidemiological Analysis and a Worldwide Haplotype Network

**DOI:** 10.3390/jof6040204

**Published:** 2020-10-03

**Authors:** Daniel Wagner C. L. Santos, Vania Aparecida Vicente, Vinicius Almir Weiss, G. Sybren de Hoog, Renata R. Gomes, Edith M. M. Batista, Sirlei Garcia Marques, Flávio de Queiroz-Telles, Arnaldo Lopes Colombo, Conceição de Maria Pedrozo e Silva de Azevedo

**Affiliations:** 1Special Mycology Laboratory—LEMI, Division of Infectious Diseases, Federal University of São Paulo, São Paulo, 04039-032 SP, Brazil; danielinfectologista@gmail.com (D.W.C.L.S.); arnaldolcolombo@gmail.com (A.L.C.); 2Division of Infectious Diseases, Federal University of São Paulo, São Paulo, 04024-002 SP, Brazil; 3Microbiology, Parasitology and Pathology Post-Graduation Program, Department of Pathology, Federal University of Paraná, Curitiba, 81531-980 PR, Brazil; vaniava63@gmail.com (V.A.V.); viniciusweiss@gmail.com (V.A.W.); sybren.deHoog@radboudumc.nl (G.S.d.H.); rrgrenata@gmail.com (R.R.G.); queiroz.telles@uol.com.br (F.d.Q.-T.); 4Bioprocess Engineering and Biotechnology Graduate Program, Federal University of Paraná, Curitiba, 81531-980 PR, Brazil; 5Center of Expertise in Mycology, Radboud University Medical Center/CWZ, 6525 GA Nijmegen, The Netherlands; 6Department of Medicine, Federal University of Maranhão, São Luís, 65080-040 MA, Brazil; edithmendonca@hotmail.com (E.M.M.B.); sirleigmarques@gmail.com (S.G.M.); 7Post-Graduation Program of Health Science, Federal University of Maranhão, São Luís, 65080-040 MA, Brazil

**Keywords:** chromoblastomycosis, Brazil, *Fonsecaea* spp., haplotype network

## Abstract

Chromoblastomycosis (CBM) is a neglected implantation mycosis prevalent in tropical climate zones, considered an occupational disease that affects impoverished rural populations. This retrospective study described clinical aspects of CBM in a hyperendemic area in Brazil and constructed a worldwide haplotype network of *Fonsecaea* spp. strains. The variables were collected from medical records using a standard report form, reporting 191 patients with CBM from Maranhão, Brazil. The mean age was 56.1 years, 168 (88%) patients were male and predominantly farmers (85.8%). The mean time of evolution of the disease until diagnosis was 9.4 years. Lower limbs (81.2%) and upper limbs (14.2%) were the main sites affected. Most patients exhibited verrucous (55%) and infiltrative plaque (48.2%). *Fonsecaea* spp. were identified in 136 cases and a haplotype network constructed with ITS sequences of 185 global strains revealed a total of 59 haplotypes exhibiting high haplotypic and low nucleotide diversities. No correlation was observed between the different haplotypes of *Fonsecaea* species and dermatological patterns, severity of disease or geographic distribution inside Maranhão. Data from this area contributed to better understanding the epidemiology of CBM. For the first time, a robust haplotype network with *Fonsecaea* strains reveals an evolutionary history with a recent population expansion.

## 1. Introduction

Chromoblastomycosis (CBM) is a one of the most prevalent implantation mycoses distributed over the tropical and subtropical regions of the world, with scattered cases reported from temperate zones. CBM, with mycetoma and phaeohyphomycosis, constitute the classic triad of implantation mycoses of major clinical importance caused by black fungi [1,2]. The disease was initially described in 1914 by Max Rudolph, a German doctor working in Brazil, with countless descriptions on all continents in subsequent decades. However, as it is not a mandatory reporting disease for health authorities in countries, the true burden of the disease remains unknown [1,3,4].

Even though it is a rarely fatal disease, CBM is an important disease entity because of its high morbidity. The disorder begins as skin-colored papules that progressively develop nodules or plaques with a scaly and warty surface, finally with tumoral (cauliflower-like), verrucous and cicatricial appearances [5,6]. More than one type of lesion can be observed in the same patient, especially in advanced stages, leading to gross disfigurement, and eventually, amputation of limbs may be required. Lymphedema, ankylosis and bacterial secondary infections are the most frequent complications, and neoplastic transformation is an important and feared complication [6,7,8].

Considering that the disease is acquired by inoculation of the fungus through the skin of contaminated plant material or soil, the farm workers, lumberjacks, vendors of farm products, animal breeders and foresters are most at risk, rendering CBM an occupational disease. CBM agents belong to a single order in the fungal kingdom, the *Chaetothyriales*, within the family *Herpotrichiellaceae*, and are found in plant thorns, soil, decaying wood and other debris [1,2,5]. Prevalent etiologic agents are *Fonsecaea* spp. (*F. pedrosoi*, *F. monophora* and *F. nubica*) in the humid and hot climate, *Cladophialophora carrionii* restricted to semi-arid areas with *Cactaceae* as main vegetation, and occasionally *Rhinocladiella* spp. (R. aquaspersa, R. tropicalis, R. similis), Phialophora verrucosa and Exophiala spp. [1,8,9]. Application of molecular tools has allowed better understanding of biodiversity and distribution of etiologic agents of CBM. As an example, while *F. monophora* is found to have a worldwide distribution, *F. pedrosoi* is typically restricted to hot and humid tropical areas, and *F. nubica* resides especially in Asia, Madagascar and occasionally in Latin America. On the other hand, the currently recognized biodiversity of CBM agents in endemic areas has clinical and epidemiological unknown implications [10,11,12].

The present study evaluates the clinical and epidemiological characteristics of CBM in a high endemic area in Brazil (Maranhão state), describing in details the patterns of skin lesions, as well as their complications. We described a large biodiversity of CBM agents in this hyperendemic area and for the first time we made a haplotypic network with 185 ITS sequences. These data allowed us to better understand their distribution within Maranhão and globally, in addition to clinical implications and population dynamics of *Fonsecaea* spp. in search of a common ancestor.

## 2. Materials and Methods

We retrospectively reviewed the medical records of all patients with chromoblastomycosis (CBM) diagnosed from 1989 to 2018 at Department of Infectious Diseases of Federal University of Maranhão (UFMA). The variables that were systematically assessed included demographic data as age, gender, occupation, comorbidities, geographical location, history of trauma, part of the body affected, symptoms, type of lesions, complications (secondary recurrent bacterial infection and neoplastic transformation), etiologic agents, treatment and outcome. Regimens of antifungal drugs therapy and physical methods were also reported.

To better describe the CBM clinical aspects and intensity of the disease we adopted the classification of Arturo Carrión (1950) and Queiroz-Telles (2009), respectively [5,13]. All patients were diagnosed with CBM through the visualization of muriform cells in skin scrapings, crusts, aspirated debris and/or tissue fragments. The direct examination of these biological samples was performed with 10–40% potassium hydroxide (KOH) and the histological analysis with Hematoxylin-eosin-stain. Additional stains (PAS and Grocott’s methenamine silver) were performed if necessary [1,2,5]. Clinical and severity classification of the disease, complications and criteria of cure (clinical, mycological, and histopathological) were defined by Queiroz-Telles [1,5].

Biological material was cultured in specific media for fungi Sabouraud Dextrose Agar (SDA, Difco Laboratories, Detroit, MI, USA) and mycobiotic agar (MA, Difco) and incubated at 28 °C, 37 °C and 42 °C for 2–3 weeks. CBM agents were identified by a classical micromorphology evaluation and molecular biology by the sequencing of ITS regions, partial genes *CDC42* (cell division cycle gene) and *BT2* (β-tubulin) [10,11,12]. The isolates were deposited at the “Microbiology Collection of the Taxonline Network—TAXon line at Federal University of Paraná, in Parana state, Curitiba Brazil (http://repositorio.utfpr.edu.br/jspui/handle/1/3332).

### Haplotype Network Analysis

For the construction of the haplotype network, we used ITS sequences of 43 clinical strains of *Fonsecaea* spp. isolates from Maranhão state and 142 other strains of *Fonsecaea* spp. distributed as follows: 53 strain sequences from other Brazilian regions (different from those isolated in Maranhão state) and 89 strain sequences from other countries recovered from GenBank (Appendix A).

The phylogenetic analysis was performed using MAFFT version 7 (multiple sequence alignment program for amino acid or nucleotide sequences—https://mafft.cbrc.jp/alignment/software/). The ITS trees were constructed with 100 bootstrap replicates using the maximum likelihood and the neighbor-joining methods, with the best evolutionary model to this dataset [14].

The haplotype network file was created using DnaSP v.5.10 software (https://ub.edu/dnasp/), which determine the extent of DNA polymorphism, such as number of polymorphic sites (S), haplotype diversity (Hd), and nucleotide diversity (π). We also tested neutrality by Tajima’s D test as well as Fu and Li’s D_ and F_test using the same program. Evolutionary relationships at the intraspecific level were evaluated using haplotype networks in order to visualize differences and diversity among isolates [15,16,17]. Median-joining networks for the dataset were obtained and visualized using the PopART (Full-Feature Software for Haplotype Network Construction) [18].

The study was approved by the local Research Ethics Committee-CEP-HUUFMA (University Hospital of the Federal University of Maranhão), according to Brazilian Resolution (Approval number: 1.276.342, approval date: 13 October 2015). All of the researchers signed a data use agreement protecting the confidentiality of the data.

## 3. Results

### 3.1. Clinical and Epidemiological Aspects

During the 30-year study period, 191 patients presenting CBM were included, all being autochthonous to Maranhão. A total of 168 were male (88%) (male/female ratio 8.3:1), with a mean age of 56.1 years (range 15–91), including 74 cases (38.7%) over 60 years old. Most patients were born and lived in warm and humid places, especially in the northern mesoregion of Maranhão (78.5%).

Patients were predominantly farmer (164, 85.8%), followed by carpenter, trucker, construction assistant, car mechanic, gardener, tradesman, and locksmith. Most of them (94.7%) came from rural areas. History of cutaneous trauma was reported by 179/191 (93.7%) patients, especially microtrauma with fragments of plants, woods, stumps or work tools, and most of patients were barefoot (84%). Data of most common comorbidities were available in 100 cases, especially diabetes mellitus (13%), hypertension (18%) and dyslipidemia (7%).

Time to diagnosis of CBM varied from 3 months to 30 years. The evolution over 10 years was observed in 66/191 (34.5%) of the cases. The most frequent location of skin lesions was lower limbs (155/191; 81.2%), followed by upper limbs (27/191; 14.2%) and face (5/191; 2.6%). Other unusual sites were trunk (3 thorax and 2 abdomen), 4 gluteus and 3 neck. Extracutaneous spread involving brain was seen in only one case. Multiple sites were affected in 12/191 (6.3%) of cases.

The most frequent clinical variety was verrucous (105/191, 55%), followed by infiltrative plaque (92/191; 48.2%), nodular (45/191; 23.6%), cicatricial (19/191; 10%) and tumorous (5/191; 2.6%). Interestingly, the ulcerated lesion pattern was observed in 20 (10.5%) patients (Figure 1). Coexistence of different patterns of lesions was observed in 128/191 (67%), reflecting a more severe pattern of disease. Patients with cutaneous lesions had no bone or joints involvements.

Regarding grades of severity, the moderate form was the most common (92/191; 48.2%), followed by severe (77/191; 40.3%) and mild (22/191; 11.5%). The mean time durations of disease at the diagnosis were 4.8, 7.2 and 10.2 years in mild, moderate and severe forms, respectively.

The most common signals and symptoms included itching (67%), peeling (46%), pain (36%), purulent discharge (35%), foul odor (29%) and edema (17%). The itching and peeling were distributed homogeneously regardless of the type and severity of lesions. Edema, odor, purulent discharge and pain prevailed in moderate and severe cases. The most frequent complication was secondary bacterial infection which occurred in 71 patients (37%).

Main demographic, clinical and epidemiological data were summarized in Table 1.

Association of CBM with other infectious diseases was observed in five patients. One of them had visceral leishmaniasis along the treatment of CBM and four patients had leprosy at the time of CBM diagnosis. Only two patients had immunosuppressive conditions at diagnosis of CBM (kidney transplantation and cutaneous lupus erythematosus). Malignant transformation of CBM lesions was described in seven cases, all diagnosed by histopathology (three well-differentiated and four poorly differentiated squamous cell carcinoma (SCC)), including six patients who developed SCC during or after treatment with itraconazole. Five patients had the affected limb amputated, one patient was cured after excision of the malignant lesion, but two patients had intra-abdominal metastasis after amputation, progressing to death. These cases were previously published by Azevedo et al. (2015) [19].

### 3.2. Laboratorial Findings

Direct mycological tests (KOH 10%) in skin biopsy fragments were performed in 172/191 (90%), with positive results in 162/172 (94.1%). Histopathological examination was performed in 183/191 (95.8%) patients, confirming the diagnosis of CBM in all cases. The main histological findings were chronic inflammatory infiltration, hyperkeratosis, granulomatous process with microabscesses and inflammatory cells showing the typically dark pigmented and thick-walled muriform cells. Skin biopsy fragments were cultured in 169 cases, but there was growth of melanized agents in 141/169 (83.4%) cases.

Conventional identification was performed in all isolates that were classified as *Fonsecaea* spp. (136 cases), *Rhinocladiella* spp. (three cases: one caused by *R.*
*aquaspersa* and two by *R. tropicalis*), *Phialophora verrucosa* (one case) and *Cyphellophora* spp. (one case). None of the agents belonged to *Cladophialophora* or *Exophiala*.

Fifty clinical isolates collected from 50 patients were identified at molecular level using multilocus sequence analyses of ITS, partial *BT2* and *CDC42* genes, allowing to classify agents as *F. pedrosoi* (45), *F. monophora* (1), *F. pugnacius* (1), *Cyphellophora ludoviensis* (1) and *R. tropicalis* (2) (Table 2).

### 3.3. Treatment

Different antifungal regimens were used in patients assisted between 1989 and 2000, particularly 5-fluorocytosine, intralesional amphotericin B, itraconazole and ketoconazole. After 2000, itraconazole was available in the public health system and became the drug of choice in the treatment of CBM.

Robust and complete treatment data were available for 73 patients, with a median follow-up period of 6.3 years (3.2 to 12 years). Forty-eight patients treated with itraconazole (200–400 mg/day) as monotherapy presented clinical and mycological cure in 6 cases (12.5%), good clinical improvement in 27 (56.2%) and 15 (31.3%) had no or insignificant clinical response.

Ten patients were treated with a combination of itraconazole (200–400 mg/day) and cryotherapy and exhibited clinical and mycological cure in 1 case (10%), good clinical improvement in 6 (60%) and 3 (30%) no or insignificant clinical improvement.

Five patients were treated with a combination of itraconazole (200–400 mg/day) and topical imiquimod 5% cream. All showed clinical improvement, but only one was cured. The association of itraconazole with topical application of amphotericin B or oral terbinafine was scarce, which did not allow us to conclude the clinical outcome. Finally, ten patients were treated exclusively with photodynamic therapy (PDT) with methylene blue as photosensitizer and a light emitting diode (LED) device as light source. All of the ten patients treated presented reduction in the compromised area after six PDT applications; however, complete healing was not achieved in any patient (Table 3).

### 3.4. Haplotype Analysis

For ITS haplotype network we employed a dataset of 185 sequences of *Fonsecaea* spp.: 43 strains from Maranhão, 53 strains from other Brazilian states and 89 genetic sequences of *Fonsecaea* spp. recovered from GenBank obtained from clinical and environmental strains (Appendix A). A total of 59 haplotypes were generated, as shown in Figure 2.

*Fonsecaea* species exhibited a total of 168 polymorphisms detected, but only 103 sites were selected for analyzes (i.e., excluding sites with gaps/missing data). In ITS region, there were 103 polymorphic segregating (S) sites, with 114 total mutations, which consisted of 81 singleton mutations and 33 parsimony-informative sites. The nucleotide (π) and haplotype diversity (Hd) were 0.099 and 0.97, respectively. The sequence data were statistically significant (*p* < 0.05) deviating from neutrality with Tajimas’s D statistic test of −2.04394. Fu and Li’s D* and F* tests also showed negative values of −4.95580 and −4.58613, respectively (*p* < 0.02), suggesting an excess of low-frequency mutations and Fu’s Fs statistic test −24.119 (Table 4).

Clinical strains of *F. pedrosoi* from Maranhão were clustered in three different haplotypes (Hp22, Hp31, Hp32), with only one strain of *Fonsecaea monophora* from Maranhão identified as haplotype 1 (Hp1). Low genetic diversity was observed, as demonstrated by the few numbers of haplotypes in clinically relevant *Fonsecaea* species (Figure 2). The only case of *F. monophora* from Maranhão was diagnosed in an endemic area for the disease, with no particularities regarding geographic and climatic aspects.

A single sequence of environmental strain of *F. monophora* isolated in Maranhão clustered in haplotype 1, which included clinical *F. monophora* strains isolated from other parts of Brazil, showing genetic similarity between clinical and environmental strains of this species. On the other hand, the only environmental strain of *F. pedrosoi* isolated in Maranhão (Hp38) did not cluster with clinical haplotypes from the same region (Hp22, Hp31 and Hp32).

No correlation was found between haplotypes of *Fonsecaea* species and their geographic distribution within Maranhão, different dermatological patterns, or disease severity.

Analysis of all *F. pedrosoi* strains from Brazil showed that they clustered into 17 haplotypes (Hp20–32, 36–38, 56) with a dominant haplotype (Hp22) that included 70 strains from all epidemic regions of the country. Sequences of *F. monophora* from Brazil clustered into 12 different haplotypes (Hp1–5, Hp10–12, 17–19, 59) and of *F. nubica* into 3 haplotypes (Hp39, Hp53, Hp57). A single isolate of *F. pugnacius* was classified as haplotype 58 (Figure 2).

Sequences strains of *F. pedrosoi* isolates of a global set of strains representing several epidemic regions resulted in 18 haplotypes (Hp33–35, Hp40–52, Hp54 and Hp55), all being distinct from those found in Brazil. The same was demonstrated for 8 haplotypes of *F. monophora* (Hp6–9, Hp13–16) found outside Brazil (Figure 2).

Haplotype network analysis of ITS sequences data of *Fonsecaea* spp. revealed four main clusters (cA~cD), largely correlated with the species of *Fonsecaea* and from geographic locations with potential recombination events (Figure 2).

## 4. Discussion

The present series represents one of the largest records of CBM cases documented in a single medical center in an endemic area of Brazil [20,21]. The state of Maranhão (Brazil), due to its geographical location between the Amazon and Northeast regions, presents several climatic patterns described in tropical areas. Most of our patients came from the Northern Mesoregion in Maranhão, where previous studies had identified at least 18 reservoirs of environmental black fungi [22,23].

Previous studies showed age distribution for patients with CBM ranging from 7 to 91 years, with most patients aging between 41 and 70 years old. In our series, the mean age at diagnosis was over 50 years, what probably is related to the long incubation period of the disease, and late diagnosis of this condition [7,8,24]. Contrasting with statistics from Venezuela, in Brazil, CBM is rarely diagnosed in younger individuals [25].

It is noteworthy that more than 80% of our casuistic of CBM was documented in males. Although there is some evidence suggesting that high estrogen levels may protect females from CBM [1,2,5], the greater infection rate of males might be secondary to differences in occupational exposures. Indeed, in Maranhao state, due to cultural peculiarities, rural workers are mainly males. Otherwise, high rates of female participation in farming have been reported especially in Venezuela, Kenya, Gabon, Taiwan, Thailand, Japan, South Africa and some provinces in Brazil and México [24,26,27,28,29,30].

Agricultural activities were reported by 86% of patients at the time of diagnosis and farming activities are strongly related to the pathogenesis of this disease, since vegetables, soils and wood are environmental sources of black fungi of CBM. Minotto, Silva and Bonifaz showed in different studies that over 70% of CBM patients were farmers [20,21,31]. It should be noted, however, that the closely related black fungus *Cladophialophora bantiana* causing brain infection and thus not being an occupational disease, also showed a preponderance of male involvement [12].

CBM shares geographic distribution with other tropical endemic and neglected diseases. Although we have reported only five cases (2.6%) of CBM association with leprosy or leishmaniasis, other authors have described a high rate of co-infection with both mentioned conditions. In addition, CBM may be also documented in combination with other fungal infections, such as paracoccidioidomycosis and mycetoma, as reported previously [6,21,32].

CBM is typically a disease of normal hosts where an association with immunosuppressive conditions is very uncommon. In the present casuistic we found only two cases of the disease in immunosuppressed hosts, including one case each of renal transplantation and systemic lupus. Other authors have also reported few cases of CBM in patients undergoing solid organ transplantation (kidney, heart and liver), pemphigus vulgaris, rheumatologic diseases (lupus erythematosus), solid tumors and chronic steroid users (type 2 leprosy reaction) [33,34].

Similar to previous publications in South America, the most common skin lesions were localized in the lower limbs, due to their proximity to the ground and the fact that rarely farmers wear appropriate personal protective equipment [20,21,24,31]. Indeed, exposed parts of both extremities are more likely to be infected by etiological agents of CBM after trauma. In this scenario, some countries as Cuba, Australia and Venezuela reports cases predominantly in the upper limbs due to the high number of workers exposed to cactus manipulation injuries, which are classic reservoirs of CBM agents [26,35,36].

Several dermatological classifications have been proposed to describe the wide clinical spectrum of CBM lesions. However, the classification introduced by Arturo Carrión in 1950 was well accepted in the clinical practice because it uses basic dermatologic descriptions, showing the nodular, warty lesions and infiltrative plaques as the most common pattern of a dermatological lesion [13]. In addition, polymorphism is predominant due to the chronic evolution of the lesions. The main symptoms are itching, scaling, pain and edema, as reported by other authors [6,7,9,21,28,30].

According to several open and non-comparative clinical trials, itraconazole (ITZ) is the standard therapy for CBM, and it is also the most commonly used antifungal drug. ITZ cure rates range from 15% to 80%. Most of our cases were treated with antifungal drugs (itraconazole) only, achieving partial clinical cure, especially in severe forms of the disease [6,26,37]. The low cure rate observed by other authors with antifungal therapy alone is due to severity of the disease, with lymphedema, excessive fibrosis, hardened tissue and low vascularization, decreasing the drug bioavailability at the site of infection. In this scenario, late diagnosis, severity of subcutaneous involvement with local barriers for drug access are all factors that may contribute to suboptimal therapeutic response [1,5,38].

Several non-chemotherapeutic methods were described for the treatment of CBM, but the prompt surgical excision is considered the best choice in case of small and initial lesions. Thermotherapy (heat and cold) and photodynamic therapy are the most used physical methods for non-initial lesions showing, generally with favorable outcome. Unfortunately, these modalities are not available in routine public services in Brazil. Our limited experience in our service obtained cure and/or clinical improvement in 7 out 10 cases treated. Several authors have shown that cryotherapy and photodynamic therapy appear promising as adjunctive therapy in patients with extensive lesions [39,40]. Recently, some Brazilian groups, including our center, have been using topical immunomodulatory drugs (imiquimod) or intramuscular administration of glucan as adjuvant therapy, obtaining significant improvement of the lesions, but without cure [41,42]. However, we clearly need more robust data generated by controlled and randomized clinical trial before defining the efficacy and safety of all those different strategies in the clinical management of patients with CBM.

The diversity of black mold species able to cause CBM is low compared to the large number of agents causing phaeohyphomycosis worlwide. Similar to other authors from Latin America (Mexico, Cuba, Jamaica and Brazil), India and Madagascar, the main etiologic agent of CBM found in our series was *F. pedrosoi*. Otherwise, our casuistic also revealed infections related to *F. monophora*, *F. pugnacius*, *Rhinocladiella aquaspersa*, *R. tropicalis*, *Phialophora verrucosa* and *Cyphellophora ludoviense*, which are uncommon agents in Brazil [6,7,21,24].

The molecular epidemiology of CBM agents has been poorly studied so far. The small number*s* of strains, as well as the lack of representativeness of the different species from all continents, do not allow reliable analysis. Ribosomal and mitochondrial DNA typing methods have been used to map the geographic origins of strains, showing differences between American, African and Asian populations of *Fonsecaea* spp. [43,44]. Najafzadeh et al. published an analysis of 81 clinical and environmental strains by amplified fragment length polymorphism (AFLP). They found that *F. pedrosoi* appeared to be of monophyletic origin and was found almost exclusively in Central and South America. On the other hand, *F. monophora* and *F. nubica* were distributed worldwide. Although these data are important in the molecular epidemiology of the main agent of CBM (*Fonsecaea* spp.), the number of strains (n = 81) was relatively small given the low distribution of these agents on the five continents. In addition, only three strains of *F. pedrosoi* from South America were included in the analysis, while most of them were from Mexico [45]. Deng et al. described genetic diversity and spatial patterns of *Fonsecaea* species from clinical and environmental samples from different parts of the globe. A total of 21 of *F. pedrosoi* (all from Latin America except 2), 30 strains of *F. monophora* (mostly from China) and 9 strains of *F. nubica* (from Latin America, China, France and Africa) were analyzed. They concluded that these agents of human disease have diverse distribution patterns and population dynamics even though they share a common ancestor within the *Chaetothyriales* order [46]. To perform a more robust analysis, Sun et al. studied 17 strains of *F. pedrosoi*, 12 of *F. monophora*, anmd 8 of *F. nubica* by multilocus analysis of five functional genes (ITS, BT2, ACT1, Lac and HmgA) and showed a separation in American, African and Asian populations [47].

The present study is the first with clinical and sequence analysis of a large number of strains (*n* = 185) of different species of *Fonsecaea* from different areas of the globe. We were able to determine the geographic distributions of the main populations of *Fonsecaea* strains inside the Brazil and around the world. The pronounced haplotype diversity (Hp1–59) found, combined with low nucleotide diversity values (π = 0.0364835) may suggest that the population is composed of related haplotypes, which is observed after a demographic event of recent population expansion. These data are corroborated by the negative value of Tajima’s test (−2.07407, *p* < 0.05), which reveals excess of recent mutations with closely related haplotypes and is supported by the Fs FU test. These negative values can be interpreted as an indication of purifying selection, or alternatively as a recent demographic expansion. Similar data have been previously reported for the genera of *Cladophialophora* and *Fonsecaea* [46,47].

The ITS alignment revealed four main clusters (cA, cB, cC and cD), representing a combination of the four main clinically relevant *Fonsecaea* species (*F. pedrosoi—*Cluster A, *F. monophora*—Cluster B, *F. nubica—*Cluster C, and *F. pugnacius*—Cluster D*)* and their global distribution (Figure 2). In Brazil, *F. monophora* was predominantly isolated in the southern and southeastern regions, where the climate is more temperate, with milder temperatures and well-defined seasons, similar to Southeast Asian climate. *Fonsecaea pedrosoi* was found throughout the Brazilian territory, especially in hot and humid climate areas, with a larger distribution in the north, northeast and mid-west. On the other hand, this species is also widely distributed in other countries in South and Central America, Africa and Southeast Asia [43,44,45,46,47].

Interestingly, environmental strains of *F. monophora* were aligned to haplotypes that contained clinical strains (Hp1), while environmental strains of *F. pedrosoi* could be aligned to haplotypes without strains isolated from humans only (Hp20, Hp38, Hp52). The degree of adaptation to the human host, or ability to grow in mammal tissue, differ between closely related species, as environmental isolation of clinical species is difficult to achieve [48]. Some mammals are hypothesized to serve as a reservoir for pathogenic species, but so far, no known animal vectors have been reported. On the other hand, it is not possible to rule out the possibility of hosting humans or other mammals may play a role in the evolution and environmental dispersion of CBM etiological agents [48].

There is a lack of correlation between clinical and epidemiological data and the different haplotypes of *Fonsecaea* spp. The only two published studies on molecular epidemiology of CBM agents do not discuss the impact of these different and diverse haplotypes on the clinical characteristics of lesions, time of evolution or severity of the disease. We tried to make this correlation, but we could not reveal any impact on the questions discussed previously. Possibly, sequencing of other genes (partial β-tubulin, *CDC42,* translation elongation factor, etc.) with a construction of a concatenated haplotype network may elucidate evolutionary data, while the genes responsible for clinical differences are yet to be found.

## 5. Conclusions

We conclude that CBM is a hyperendemic disease in Maranhão state that compromises especially male individuals after 50 years of age who work in agricultural settings. As it is a chronic disease with low mortality, medical assistance is often postponed, allowing patients to reach the health service in a late stage with complications, such as lymphedema, bacterial infections and sometimes neoplastic transformations. Verrucous and infiltrative plaque are the most common dermatological patterns in our center. *Fonsecaea pedrosoi* is the main agent, while *F. monophora*, *Rhinocladiella* spp. and others rarely described in our study. Although the haplotype analysis concluded that *Fonsecaea* spp. have a high haplotypic diversity and low nucleotide diversity suggesting a demographic event of recent population expansion, we were able to characterize very divergent haplotypes of *F. pedrosoi* and *F. monophora* distributed in Latin America and Asia.

## Figures and Tables

**Figure 1 jof-06-00204-f001:**
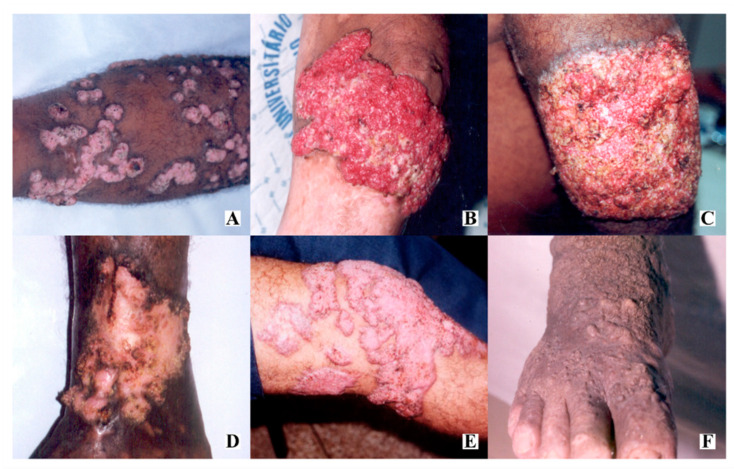
Pattern of dermatological lesions of chromoblastomycosis: (**A**) nodular lesions, sometimes confluent; (**B**) tumoral and hypervascularized lesion with spontaneous bleeding; (**C**) warty plaque; (**D**) scar lesion with central healing areas; (**E**) polymorphic lesion, with nodules, plaques and tumor; (**F**) verruciform lesion.

**Figure 2 jof-06-00204-f002:**
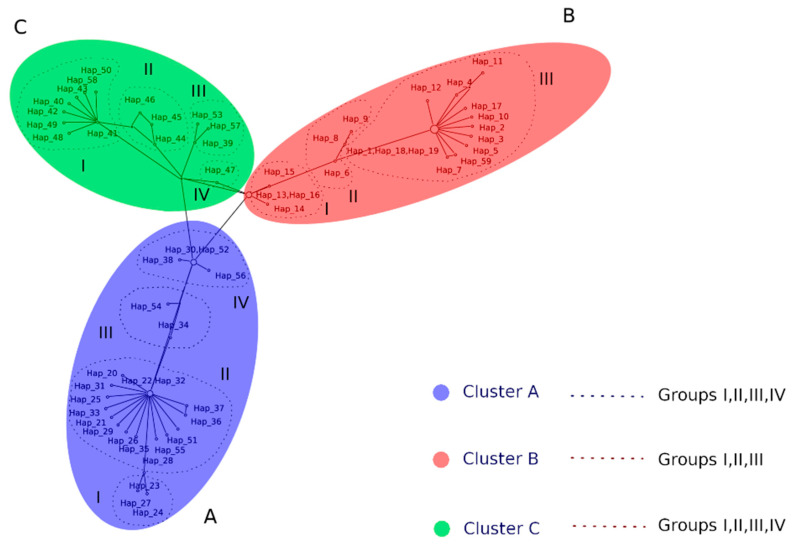
Internal Transcribed Spacer (ITS) haplotype network obtained from dataset of 185 sequences of *Fonsecaea* spp. and the 59 haplotypes generated. Cluster **A**—*F. pedrosoi*: shared in 4 groups (Group I: Hp23,24,27 obtained from Brazil; Group II: Hp20–22,25,26,28,29,31–33,35–37,51,55 obtained from Brazil, Argentina, Uruguay, Venezuela, Mexico and Lybia; Group III: Hp34,54 obtained from Puerto Rico; Group IV: Hp30,38,52,56 obtained from Brazil and Venezuela); Cluster **B**—*F. monophora*: shared in 3 groups (Group I: Hp13-16 obtained from China and Malaysia; Group II: Hp6,8,9 obtained from China, India and Guinea; Group III: Hp1–5,7,10–12,17–19,59 obtained from Brazil, China, USA, Cuba, Poland); Cluster **C**—*F. nubica*: shared in 4 groups (Group I: Hp40–43,48–50 obtained from China; Group II: Hp44–46 obtained from China, Suriname, Camaroon; Group III:39,53,57 obtained from Brazil; Group IV: Hp47 obtained from China).

**Table 1 jof-06-00204-t001:** Demographic, clinical and epidemiological data of 191 CBM cases from Maranhão state, Brazil.

Variables	N (%)
Male/Female	168 (88%)/23 (12%)
Age strata (years) at diagnosis	
10–19	1
20–29	6
30–39	13
40–49	28
50–59	70
60–69	48
70–79	22
>80	3
Mesoregions in Maranhão	
North	150 (78.5%)
West	19 (10%)
East	10 (5.2%)
Central	12 (6.3%)
Environmental exposition	
Living in rural areas	181 (94.7%)
Farmers	164 (85.8%)
Trauma	179 (93.7%)
Mean time in years of evolution of the disease until diagnosis of CBM	9.4 (3m-30y)
Patterns of dermatological lesion	
Verrucous	105/191 (55%)
Plaque	92/191 (48.2%)
Nodular	45/191 (23.6%)
Tumorous	5/191 (2.6%)
Cicatricial	19/191 (10%)
Ulcer	20 (10.5%)
Mixed form	128/191 (67%)

**Table 2 jof-06-00204-t002:** Etiologic agents of CBM (Maranhão state, Brazil) identified by conventional and molecular tools.

Etiological Agent and Identification Method	N
Conventional identification	141
*Fonsecaea* spp.	136
*Rhinocladiella* spp.	2
*Rhinocladiella aquaspersa*	1
*Phialophora verrucosa*	1
*Cyphellophora* spp.	1
Molecular identification	50
*Fonsecaea pedrosoi*	45/50
*F. monophora*	1/50
*F. pugnacius*	1/50
*Rhinocladiella tropicalis*	2/50
*Cyphellophora ludovicensis*	1/50

**Table 3 jof-06-00204-t003:** Clinical efficacy of different therapeutic strategies used in 73 patients with CBM in from Maranhão state, Brazil.

Therapeutic Strategy (Patients with Follow-Up)	N = 73 (%)
Itraconazole	48
Cure	6 (12.5%)
Clinical Improvement	27 (56.2%)
No or insignificant improvement	15 (31.3%)
Itraconazole and cryotherapy	10
Cure	1 (10%)
Clinical Improvement	6 (60%)
No or insignificant improvement	3 (30%)
Itraconazole and topical imiquimod 5%	5
Clinical Improvement	5 (100%)
Photodynamic therapy	10
Clinical Improvement	10 (100%)

**Table 4 jof-06-00204-t004:** Diversity and molecular evolutionary parameters of 185 sequences of *Fonsecaea* spp. using ITS region as a phylogenetic marker.

Variables	N
Number of sequences	185
Number of sites	168
Number of sites (excluding sites with gaps/missing data)	103
Analysis of DNA polymorphism	
Number of mutations, Eta	114
Number of singleton mutations, Eta(s)	81
Number of haplotypes	59
Nucleotide (π)	0.099
Haplotype diversity (Hd)	0.97
Variance of haplotype diversity	0.00018
Neutrality analysis	
Number of segregating sites analyzed	94
Fu’s Fs statistic	–24.119
Fu and Li’s D* statistic	–4.95580
Fu and Li’s F* statistic	–4.58613

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
