# Peer review of "Chromoblastomycosis in an Endemic Area of Brazil: A Clinical-Epidemiological Analysis and a Worldwide Haplotype Network"

_jof, 2020, doi:10.3390/jof6040204_

Round 1

Reviewer 1 Report

Dear Authors,

The manuscript ID: jof-940208 named: “Chromoblastomycosis in an endemic area of Brazil: a clinical-epidemiological analysis and a worldwide haplotype network” written by Daniel Wagner C. L. Santos, Vania Aparecida Vicente, Vinicius Almir Weiss, G. Sybren de Hoog, Renata R. Gomes, Edith M. M. Batista, Sirlei Garcia Marques, Flávio de Queiroz-Telles, Arnaldo Lopes Colombo and Conceição de Maria Pedrozo e Silva de Azevedo is worth to be published.

Although chromoblastomycosis has been known for 100 years, it still remains an unresolved clinical problem. Relapses are frequently reported. Also, many issues regarding aetiological agents, their appearance in the environment, geographical distribution and pathogenicity need to be addressed. Host genetic susceptibility to the disease and the immune system response mechanism should be also considered.

I think, it is original article. The presented results are based on extensive biological studies (191 patients with CBM from endemic area of Brazil, clinical and epidemiological studies and haplotype network analysis). The results are documented, showed in the form of figures and tables and properly interpreted. The ITS world haplotype network of Fonsecaea spp. strains and their global distribution is particularly interesting. This manuscript is well-written and correctly prepared. The whole article (Introduction, Material and Methods, Results, Discussion and Conclusion) is appropriately organized. Introduction is concrete. Adequate methods were used to perform these studies. The Authors presented very interesting conclusions. These data contributed to better understanding the epidemiology of CBM. Moreover, a robust haplotype network with Fonsecaea strains reveals an evolutionary history with a recent population expansion.

I have some minor comments for improving the manuscript:

References: should be prepared according to the instructions for Authors

Line 55: spp. – spp. (without italics)

Line 55: and – (without italics)

Line 55: spp. [1,8,9]. – (without italics)

Line 321: small numbers – please standardize the font

According to me, this manuscript is valuable and may be accepted for the publication in “Journal of Fungi”.

With highest regards,

Author Response

Dear Editor,
First of all, thank you so much for your time, support and all helpful suggestions regarding our paper entitled “Chromoblastomycosis in an endemic area of Brazil: a clinical-epidemiological analysis and a worldwide haplotype network”.

In this revised version of our manuscript we incorporated all reviewer´s suggestions, including the adjustments in references, standardization of font and corrections of abbreviations or words in italics.

Please find bellow our point by point answers to the reviewer’s comments.

Reviewer 1

General considerations: Although chromoblastomycosis has been known for 100 years, it still remains an unresolved clinical problem. Relapses are frequently reported. Also, many issues regarding aetiological agents, their appearance in the environment, geographical distribution and pathogenicity need to be addressed. Host genetic susceptibility to the disease and the immune system response mechanism should be also considered. I think, it is original article. The presented results are based on extensive biological studies (191 patients with CBM from endemic area of Brazil, clinical and epidemiological studies and haplotype network analysis). The results are documented, showed in the form of figures and tables and properly interpreted. The ITS world haplotype network of Fonsecaea spp. strains and their global distribution is particularly interesting. This manuscript is well-written and correctly prepared. The whole article (Introduction, Material and Methods, Results, Discussion and Conclusion) is appropriately organized. Introduction is concrete. Adequate methods were used to perform these studies. The Authors presented very interesting conclusions. These data contributed to better understanding the epidemiology of CBM. Moreover, a robust haplotype network with Fonsecaea strains reveals an evolutionary history with a recent population expansion.

Comment: References should be prepared according to the instructions for Authors

Author's reply: Done. We incorporated all the adjustments of all references to the model recommended by the journal. By the way, references 22 and 23 (line 465 - 472) were duplicated. We did the correction and replaced the reference 23 for an article consistent with the text [Garcia Marques, S.; Pedroso E Silva, C.M.; Aparecida Resende, M.; Moura Silva, A.A.; Mendes Caldas, A.J.; Lopes Costa, J.M. Detection of delayed hypersensitivity to Fonsecaea pedrosoi metabolic antigen (chromomycin). Nihon Ishinkin Gakkai Zasshi 2008,49(2),95-101]. The corrections were highlighted in yellow.

Comment: Line 55: spp. – spp. (without italics)

Author's reply: Done

Comment: Line 55: and – (without italics)

Author's reply: Done

Comment:

Line 55: spp. [1,8,9]. – (without italics)

Author's reply: Done

Comment:

Line 321: small numbers – please standardize the font

Author's reply: Done. After the corrections requested by the reviewer 2, this adjustment was replaced in line 326 (no more at 321). The correction was highlighted in yellow.

Conceição de Maria Pedrozo e Silva de Azevedo

Department of Medicine, Federal University of Maranhão, São Luís, MA, Brazil

Post-graduation Program of Health Science, Federal University of Maranhão, São Luís, MA, Brazil

Correspondence: conceicaoposgradufma@gmail.com

Reviewer 2 Report

Evaluation and comments to the manuscript ID jof-940208, and entitled “Chromoblastomycosis in an endemic area of Brazil: a clinical-epidemiological analysis and a worldwide haplotype network.”.

Authors: Daniel Wagner Santos et al.

Overall, the manuscript is written in correct scientific language. I did not notice any major factual or editorial errors at work, and I do not mention any minor errors, because they are in some sense something normal in this type of work and they do not affect the general perception of work.

In my opinion, the experimental design and data analysis are appropriate and the introduction is correct. I think that we can always do something better, but this manuscript is on a good level. However, a few things need to be corrected in the manuscript before it is accepted for printing:

1) It is necessary to correct minor omissions, for example by removing italics in the notation of commas, expressions "spp." etc.

2) Fig. 1 –  where are “a” and “d” in fig.?

3) Fig. 2 – improve readability and quality

4) Tables – columns must have names.

Author Response

Dear Editor,
First of all, thank you so much for your time, support and all helpful suggestions regarding our paper entitled “Chromoblastomycosis in an endemic area of Brazil: a clinical-epidemiological analysis and a worldwide haplotype network”.

In this revised version of our manuscript we incorporated all reviewer´s suggestions, including the adjustments in references, standardization of font and corrections of abbreviations or words in italics.

Please find bellow our point by point answers to the reviewer’s comments.

Reviewer 2

Evaluation and comments to the manuscript ID jof-940208, and entitled “Chromoblastomycosis in an endemic area of Brazil: a clinical-epidemiological analysis and a worldwide haplotype network.”.

Authors: Daniel Wagner Santos et al.

Overall, the manuscript is written in correct scientific language. I did not notice any major factual or editorial errors at work, and I do not mention any minor errors, because they are in some sense something normal in this type of work and they do not affect the general perception of work.

In my opinion, the experimental design and data analysis are appropriate and the introduction is correct. I think that we can always do something better, but this manuscript is on a good level. However, a few things need to be corrected in the manuscript before it is accepted for printing:

Comment: It is necessary to correct minor omissions, for example by removing italics in the notation of commas, expressions "spp." etc.

Author's reply: Done. We incorporated all the adjustments (Lines 51, 53, 55, 189, 324,326). The corrections were highlighted in yellow.

Comment: Fig. 1 à where are “a” and “d” in figure 1?

Author's reply: Done. We incorporate the missing letters (“a” and “d”) in Figure 1 and the quality has been improved. The new “Figure 1” has been incorporated into the main text and is attached to the upload.

Comment: Fig. 2 à improve readability and quality

Author's reply: Done. We have incorporated changes to the legend of “Figure 2” and the quality / readability have been improved. We kept the name of Cluster A / B / C, however we use Roman numbers in the subdivisions within each Cluster. The new “Figure 2” has been incorporated into the main text and is attached to the upload (Linea 213-222).

Comment: Tables – columns must have names

Author's reply: Done. We incorporate names for all table columns. Some tables have been reconfigured. The corrections were highlighted in yellow.

Hopefully the paper may now be considered suitable for publication.

Yours Sincerely,

Conceição de Maria Pedrozo e Silva de Azevedo

Department of Medicine, Federal University of Maranhão, São Luís, MA, Brazil

Post-graduation Program of Health Science, Federal University of Maranhão, São Luís, MA, Brazil

Correspondence: conceicaoposgradufma@gmail.com